# Effects of Field Aging on Material Properties and Rutting Performance of Asphalt Pavement

**DOI:** 10.3390/ma16010225

**Published:** 2022-12-26

**Authors:** Haoyang Wang, Yu Zhu, Weiguang Zhang, Shihui Shen, Shenghua Wu, Louay N. Mohammad, Xuhui She

**Affiliations:** 1RoadMaint Co., Ltd., Beijing 100095, China; 2Research Institute of Highway, Ministry of Transportation, Beijing 100008, China; 3School of Transportation Engineering, Southeast University, Nanjing 210096, China; 4Rail Transportation Engineering, Pennsylvania State University, 216E Penn Building, Altoona, PA 16601, USA; 5Department of Civil, Coastal, and Environmental Engineering, University of South Alabama, Baton Rouge, AL 36688, USA; 6Louisiana Transportation Research Center, Department of Civil and Environmental Engineering, Louisiana State University, Mobile, LA 70808, USA

**Keywords:** asphalt aging, material property, climate condition, pavement structure, field rut depth

## Abstract

This study evaluates field asphalt aging based on material property changes in pavement with time, and investigates if such changes could have an impact on field rutting performance. Four projects from three different climate zones were monitored as part of the NCHRP 9–49A project at two stages: during pavement construction and two to three years after opening it to traffic. Construction information were collected, and field cores were drilled at both stages to evaluate the material properties of recovered asphalt binder and asphalt mixture. Field rut depth was also measured. In addition, pavement structure, climate and base/subgrade modulus information were also obtained. Results indicate that the asphalt mixture stiffening is caused in major part by asphalt aging. However, the effect of asphalt aging on pavement mixture property may not follow a proportional liner trend. The parameters that are most sensitive to field ageing are MSCR *R3.2* and dynamic modulus. It is also found that the variables which showed a good ranking trend with the field rut depth are climate condition (relative humidity, high temperature hour, solar radiation), material properties (Hamburg rut depth, rutting resistance index, high temperature performance grade, MSCR, and dynamic modulus, base and subgrade moduli), as well as air voids.

## 1. Introduction

Asphalt binder is a material widely used in paving engineering to hold aggregates. It is a special hydrocarbon composite with complex mechanical properties and plays a major role in determining the performance of asphalt pavements. However, asphalt materials tend to oxidize under the influence of environmental factors such as temperature, humidity and solar radiation, leading to changes in its mechanical and rheological properties.

Laboratory and field studies have shown that after oxidization, the stiffness of the asphalt binder increased [1,2,3]. Previous research using recovered binder from field cores found that high-temperature performance grade (PG) increased 2.4 to 26.6 °C for pavements aged between 10 and 82 months [4]. With ageing, the weak attractions of the asphaltene are destroyed, and asphalt molecules change their orientation and become more tightly packed [5]. In addition, the ageing process resulted in stronger associations between asphalt components due to the generation of polar carbonyl groups, which increased the asphaltene fraction [2]. Changes in both its elastic modulus and its viscosity lead to stiffening of the binder.

For the asphalt mixture, numerous laboratory studies have been implemented to investigate property changes with the aging of the asphalt binder. They have shown that the resilient modulus of long-term oven-aged specimens is 50–100% higher than that of short-term oven-aged specimens [2,6]. Moreover, some researchers assumed that due to diffusion resistance, binder oxidation rates in mixtures are slower than that in thin-film neat binders due to diffusion resistance [7,8,9,10]. The Hirsch model is often used to describe the relationship between asphalt and the asphalt mixture; however, it was found that due to binder hardening, the mixture ageing actually stiffened the mixture more than the model predicted.

However, few field tests have been performed due to the impact of multiple factors such as climate and binder modification. The current research only focuses on the influence of the dynamic modulus. For instance, the dynamic modulus of field cores in Texas increased by approximately 70% and 130% at the end of 8 and 14 months compared with that at 1 month [3]. On the other hand, such increases are much less, with 27–31% after one year of service in the cold regions of Sweden [11]. Furthermore, it was found that polymer- and rubber-modified binders may reduce the ageing rate in contrast to unmodified binders [4,12,13], and the Evotherm WMA binder generally ages faster than the HMA and Foaming WMA binders [1,3,4].

In summary, most aging studies are focused on laboratory aging, and very few studies have attempted to examine the effects of field aging on the property evolution of asphalt and asphalt mixture. In addition, the field studies in most cases were based on one specific project; thus, the effect of factors such as climate condition, HMA layer thickness and binder types cannot be well analyzed [14,15]. More importantly, limited research exists on how the change in field binder properties over time has affected field mixture properties [16,17]. This study evaluates and quantifies the effect of field aging on the rutting performance of asphalt binders and asphalt mixture over time using field projects from different climatic zones. The study also analyzes the effect of asphalt binder property change on mixture alternation.

## 2. Methodology

Four projects selected from the NCHRP 9–49A project report 843 were used for performing the analysis introduced in this paper. These projects were selected to cover different pavement structures and material properties. Within each project, field cores were taken and material properties that are typically used to describe asphalt stiffness and rutting resistance were tested, consisting of binder high temperature PG, binder multiple stress creep recovery (MSCR), mixture dynamic modulus and mixture Hamburg rut depth. These material properties were tested based on field cores taken at two different periods. In the first series, cores were obtained shortly after pavement construction and in the second series cores were obtained two to three years after the construction was finished. All the field cores were taken inside the wheel path to study purely the effect of aging to exclude the effects of traffic load. Other factors that may affect field aging and field rut depth were also collected, consisting of in-place air voids, pavement structure and climate. Additionally, aggregate gradation and asphalt content were obtained to confirm that no significant changes in the cores happened due to traffic. The material properties between the two series of field aging were compared. The effect of asphalt property change on the asphalt mixture properties was also analyzed.

## 3. Project Information

Four projects that were constructed in 2011 and 2012 were investigated, which covered different climatic zones, pavement types (HMA and WMA), traffic levels and pavement structures. The four pavement projects were located in four states: Montana, Tennessee, Iowa and Louisiana, and are therefore referred to the MT I-15, TN SR 125, IA US 34 and LA US 61. Three 61-m test sections of HMA and WMA pavements were selected for further study. Field construction information of the four projects that was collected includes:Pre-construction information: mixture design, technology type (i.e., HMA or WMA), existing pavement structure, existing pavement conditions, target mixing and compaction temperatures, mile post or GPS information for the selected three 61 m research test sections, etc.During-construction information: plant modification, weather, material type, aggregate moisture content, mixing and compaction temperatures, transportation distance, in situ density, procurement of field gyratory-compacted samples, loose mixtures and raw materials, and any other significant information that should be noted, etc.Post-construction information: quality control/quality assurance (QC/QA) data, procurement of cores, falling weight deflectometer (FWD) testing, location of field cores, annual average daily truck traffic (AADTT).

The key information for the four field projects is concluded in Table 1. Each project includes at least one WMA technology and a homological control HMA pavement. More detailed information of the four projects can be found in the NCHPR report 843 [18].

### 3.1. Pavement Structure

The pavement structure that was used for each project is shown in Figure 1. The thickness of each pavement layer is presented, and the subgrade soil is assumed to be infinite in depth. Two existing pavement structure types were selected, including flexible pavements and a combination of asphalt/PCC pavements.

### 3.2. Field Climate Information

Climate information, consisting of high temperature hour, shortwave solar radiation and humidity are highly related to asphalt aging [2,18,19] and were obtained from the long-term pavement performance (LTPP) website, InfoPave. The number of air temperature hours referred to in this paper is the hours when the pavement temperatures are higher than 25 °C, as recommended [19]. As shown in Figure 2, the MT I-15 project has the lowest values of relative humidity, air temperature hour > 25 °C, shortwave solar radiation and indicating the least ageing impact. The other three projects experienced similar relative humidity and shortwave solar radiation. The LA US 61 project had the highest value of air temperature hour > 25 °C and may have encountered the most severe ageing. Note that the climate information comprises the accumulated values which cover the period of being open to traffic until the second round of field core samples were taken.

## 4. Data Collection

### 4.1. Field Climate Information

Cores were taken in the field from the non-wheel path to limit the potential of pre-existing damage in the material. In the laboratory, the existing pavement was removed for further core tests. After the core fracture test, binder extraction and recovery were performed using the entire overlay specimen to evaluate the average aging effect without taking the aging gradient variation through overlay thickness into account.

### 4.2. Binder Extraction and Recovery

Asphalt binders were extracted based on AASHTO T164 and recovered according to AASHTO R59. The used chemical was a combination of 85% Toluene and 15% Ethanol by volume. Both WMA and HMA on-site cores were heated 110 °C until they were loose enough to separate. The separated mixtures were cooled down at room temperature before extraction. The minimum mass of samples used for binder extraction was determined by nominal maximum aggregate size (NMAS). Usually, several extractions were needed until the extract was no darker than a light straw color. Recovered binders were tested by taking them as short-term aged (rolling thin-film oven-aged) asphalt, as suggested [14,15].

### 4.3. Aggregate Gradation and Asphalt Content

Aggregate gradation and asphalt content were checked using the field cores after fracture tests for both the first-round and the second-round surveys. The asphalt content was determined in accordance with AASHTO T 308, and aggregated gradation was checked following AASHTO T 30.

### 4.4. Material Properties

In the laboratory, an overlay of field cores and recovered asphalt binders was used to conduct a series of laboratory tests to determine the physical and engineering material properties. Table 2 shows the summary of all the laboratory mixture and binder tests. The MSCR test temperatures were determined based on the high pavement temperature of specific project locations obtained from LTPPBind Version 3.1 software (Federal Highway Administration, Washington, DC, USA) at 98% reliability.

## 5. Results and Analysis

### 5.1. Air Voids

Air voids are defined as the pockets of air between the asphalt-coated aggregate particles in a compacted asphalt paving mixture. Asphalt mixtures of high air voids generally show a faster rate of asphalt aging than the mixture with low air voids [2], which may be due to the fact that asphalt oxidation is the chemical reaction of asphalt with oxygen [5], and high air voids increased the contact areas between air and asphalt mixture.

Figure 3 summarizes the average air voids content based on three core replicates. Error bars that represent standard deviation are also shown. As it can be seen, the air voids from the first round exhibited similar values in general to the second round for most projects (except for MT I-15). This is within our expectations since all the cores were taken at the non-wheel path (with limited traffic load) and the differences in the air voids between the two rounds could be caused by construction variation. For the MT I-15 project, chip seal was placed one year after being open to traffic, which could affect the air voids from the second round.

### 5.2. Aggregate Gradation and Asphalt Content

Aggregate gradation is expressed by percent passing of coarse aggregate (aggregate predominately retained on 4.75 mm sieve) and fine aggregate (aggregate almost entirely passing the 4.75 mm sieve). Aggregate gradation helps determining important asphalt pavement properties such as stiffness and rutting resistance. The asphalt content is the ratio between the asphalt weight and the total mixture weight (asphalt plus aggregate). Higher asphalt content generates thicker film thickness and reduces aging effect, whereas lower asphalt content in general increase mixture stiffness and result in better rutting resistance [2].

Table 3 summarizes aggregate gradation determined based on field cores. Note that the aggregate gradation shows a range because each project concludes two or more HMA and WMA pavements; this range covers all the pavements from specific project. As observed, there is no significant gradation difference between the two rounds for all the projects except for the IA US 34. A large percentage of the aggregate in the IA US 34 project is limestone, which has a lower strength and is weaker than the other paving aggregate types such as granite and gravel.

Asphalt content was also determined based on field cores. The asphalt content difference between the two rounds is smaller than 0.1%; such a small difference should not significantly affect the rutting resistance of asphalt pavement.

### 5.3. Recovered Binder High-Temperature PG

The high-temperature PG evaluates the rheology properties of the binder at various temperatures. Typically, a greater high-temperature PG value indicates a stiffer asphalt binder (more rutting resistance) caused by aging or asphalt modification.

Figure 4 illustrates the high-temperature PG comparisons between the first and the second rounds of the extracted binders. For projects examining TN SR 125, IA US 34, and LA US 61, as shown, the high-temperature PG of the second round of extracted binders is always higher than those of the first round, which implies a clear effect of field aging during the two to three years of service.

For the MT I-15 project, it was observed that the PGs of the extracted binders in the second round were close to those in the first round. The cold local climate as shown in Figure 3, plus the chip seals that covered the surfaces, could have reduced the aging effect of the asphalt. The binders of the TN SR 125 project show the highest PG increase over time, which may be explained by the thin asphalt layer applied. The IA US 34 binders have a relatively small high-temperature PG, which indicates less rutting resistance ability.

For the LA US 61 project, the high-temperature PG increase is relatively small, considering that the high temperature hour > 25 °C is the highest among the four projects. Such slow ageing could be caused by the PG 76-XX polymer-modified asphalt (PMA) binder used. Based on on-site cores from more than 20 field projects, Zhang et al. [4] found that the PG76-XX PMA binders generally aged less than PG64-XX and PG70-XX binders. The reason for less ageing may be that polymers can prevent the formation of sulfoxides on ageing [20].

### 5.4. Recovered Binder MSCR Test

The MSCR test is used to evaluate the asphalt binder’s potential for permanent deformation. The test is performed by dynamic shear rheometer (DSR) under shear creep and recovery at two stress levels (0.1 and 3.2 kPa) at a specified temperature. The creep portion of the test lasts for 1 s at a constant stress, which is followed by a 9 s recovery. Ten creep and recovery cycles were tested at each stress level. Two parameters, non-recoverable creep compliance (Jnr) and percent recovery (R), were obtained from the test and can be calculated using:(1)$$\in_{r}\left(3.2,N\right)=\frac{\left(\in_{1}-\in_{10}\right)\times 100}{\in_{1}}$$
(2)$$R_{3.2}=\frac{SUM\left[\in_{r}\left(3.2,N\right)\right]}{10}\;for\;N=1\;to\;10$$
(3)$$J_{nr}\left(3.2,N\right)=\frac{\in_{10}}{3.2}\;\;\;\;\;\;\;\;\;\;\;\;\;\;\;\;\;\;\;$$
(4)$$J_{nr_{3.2}}=\frac{SUM\left[J_{nr}\left(3.2,N\right)\right]}{10}\;for\;N=1\;to\;10$$
where ∈_1_ is the strain value at the end of the creep portion (i.e., after 1 s) of each cycle, and ∈_10_ is the strain value at the end of the recovery portion (i.e., after 10 s) of each cycle.

This paper aims at 3.2 kPa stress level since high stress levels are more important for rut depth development. ∈*_r_* (3.2, N) and *Jnr* (3.2, N) indicate percent recovery and nonrecoverable creep compliance at each cycle, respectively. Finally, average percent recovery (*R3.2*) and average nonrecoverable creep compliance (*Jnr3.2*) at 3.2 kPa were obtained by calculating the mean percent recovery and mean nonrecoverable creep compliance from cycles 1 to 10.

Figure 5a,b compare the *Jnr3.2* and *R3.2* between the first- and the second-round-extracted binders, respectively. For the MT I-15 project, no obvious changes in *Jnr3.2* and *R3.2* were observed between the two rounds, which again could be ascribed to the reduced aging effect due to the chip seal surface treatment. For the other projects, both HMA and WMA binders from the second-round cores show lower *Jnr3.2* and higher *R3.2* values than those in the first round, indicating an improved rutting resistance of the pavements.

Among the four projects, the IA US 34 project shows the highest *Jnr3.2* and the lowest *R3.2*, which implies the high potential of rutting susceptibility. Asphalt binders from the MT I-15 project show the lowest *Jnr3.2* and the highest *R3.2* values, illustrating good rutting resistance. The *Jnr3.2* and *R3.2* values between the TN SR 125 and the LA US 61 are similar to each other.

### 5.5. Field Core Dynamic Modulus

The dynamic modulus is defined as the complex modulus absolute value calculated by dividing the peak-to-peak stress by peak-to-peak strain for a material of a sinusoidal loading on a material. The dynamic modulus is a performance-related property that can evaluate the mixture and characterize the stiffness of asphalt mixtures for mechanistic–empirical pavement design.

The indirect tension dynamic modulus test was conducted to determine the dynamic modulus while considering the limitations of the core geometry. A sinusoidal compressive loading was applied to the diametric axis of an unconfined cylindrical test specimen. Test temperatures and frequencies are shown in Table 2. The loading was applied on each sample to achieve the target strain levels (40–60 horizontal microstrain and < 100 vertical microstrain) in the linear viscoelastic region [21,22,23]. The load–deformation mathematical relationship in the indirect tension-loading mode is given by:(5)$$\left|E^{*}\right|=\frac{2P_{0}}{\pi ad}\frac{\beta_{1}\gamma_{2}-\beta_{2}\gamma_{1}}{\gamma_{2}V_{0}-\beta_{2}U_{0}}\;\;\;\;\;\;\;\;\;\;\;$$
where *P*_0_ is the peak-to-peak load in N, a indicates loading strip width measured in meters, *d* means the thickness of specimen in meters and *V*_0_ and *U*_0_ represent peak-to-peak vertical deformation and peak-to-peak horizontal deformation in meters, respectively. *γ*_1_, *γ*_2_, *β*_1_, and *β*_2_ are geometric constants.

Figure 6 presents the dynamic modulus test results at a test temperature of 30 °C and a test frequency 0.1 Hz since the asphalt pavement is more susceptible to rutting under relative high temperatures and a low frequency [24]. It is observed in the figure that most projects show an obviously higher dynamic modulus of the second-round field cores than that from the first-round. Since all the cores were taken from the non-wheel path, this dynamic modulus increase is presumably due to the significant field aging of the asphalt binder. The MT I-15 project shows the least increase in the dynamic modulus, which could be affected by the chip seals applied. The high amount of asphalt used in chip seal fills voids of overlay and may increase film thickness and asphalt content, thereby reducing field aging and resulting in a similar dynamic modulus between the two rounds.

It is also seen that the field aging increases dynamic modulus up to 800% in the second round for IA US 34 HMA compared with the value from the first round. Meantime, the high temperature PG and the MSCR *Jnr3.2* and *R3.2* of the same pavement changed by 12.8%, 69% and 165%, respectively. This finding denotes that the increase rates of the asphalt binder and the dynamic modulus are not the same, which will be discussed below.

### 5.6. Field Core HWT Test Results

The HWT is a widely used test method to determine the rutting resistance and moisture susceptibility of asphalt mixture due to weakness in the aggregate structure, inadequate binder stiffness, or moisture damage. This method measures the rut depth and number of passes to failure and provides information about the rate of permanent deformation from a moving, concentrated load.

The HWT was performed following AASHTO T 324. All tests were conducted at a temperature of 50 °C under wet conditions. The speed of the wheel was set as 52 passes per minute. The test terminated when either the rut depth achieved 12.5 mm or a pass number of 20,000 was reached.

Figure 7 summarizes the HWT rut depth at 10,000 passes. This pass number was selected because all the first-round cores from the IA US 34 project reached the test threshold value of 12.5 mm at 10,000 passes. It is observed that in general, the rutting resistance of the second-round core is higher than that from the first round, except for the MT I-15 in which the chip seal may have reduced the aging effect. Since the aggregate gradation and asphalt content between the two rounds did not change greatly, the improved rutting resistance should have been contributed to to a major extent by asphalt aging. Bonding between asphalt and aggregate particles provide significant force in resisting mixture to deform, less aged asphalt is more flow and provides a slip plan between aggregates which facilitates aggregate movement and mixture is easy to deform. In contrast, flow conditions of aged asphalt reduced due to increased viscosity, which helped produce better adhesion between aggregates, and aggregate movement became more restricted. In this case, deformation lessened.

It is also seen that the field aging increases the dynamic modulus up to 800% in the second round for IA US 34 HMA compared with the value from the first round. Meanwhile, the high-temperature PG and the MSCR *Jnr3.2* and *R3.2* of the same pavement changed by 12.8%, 69% and 165%, respectively. This finding denotes that the increase rates of the asphalt binder and the dynamic modulus are not the same, which will be discussed below.

### 5.7. Effect of Asphalt Aging on Mixture Properties

In order to study the effects of the variation in asphalt binder properties due to aging on material properties of asphalt mixture, changes to them between the first and second rounds were calculated. These are the increase in high-temperature PG, decrease in MSCR *Jnr3.2*, increase in MSCR *R3.2*, increase in dynamic modulus and decrease in HWT rut depth, respectively.

As seen in Figure 8, the asphalt property changes with dynamic modulus increase correlate well with ageing. The dynamic modulus values increased with the increase in high-temperature PG and MSCR *R3.2*, and the decrease in MSCR *Jnr3.2*. The magnitude changes in the binder properties between the two rounds are 1.2–8.4 °C for high-temperature PG, 0.1 to 24.6% for MSCR *R3.2* and 0.02 to 1.07 kPa for MSCR *Jnr3.2*. Those changes corresponded to an increase in dynamic modulus up to 1356 MPa.

Figure 9 shows the relationship between the binder property change and HWT rut depth. As noted, there was a loose relationship denoting that rut depth in general increased with the increase in high-temperature PG and decrease in MSCR *Jnr3.2*. No correlation between HWT rut depth and MSCR *R3.2* over time was found. As it can be seen, the increase in high-temperature PG (1.2 to 8.4 °C) and the decrease in MSCR *Jnr3.2* (0.02 to 1.07 kPa) reduced the HWT rut depth to the maximum value of 7.3 mm between the two rounds.

Additionally, the percentage increase/decrease in each material property between the two rounds was calculated and shown in Figure 10. As it can be seen, regarding binder properties, the high-temperature PG has the least percentage increase with the maximum value of 32.2%, whereas the MSCR *R3.2* experienced the most increase with the maximum value of 325%. As for the mixture, the dynamic modulus and the HWT rut depth increased up to 673.1% and 64%, respectively. The large range of MSCR *R3.2* and mixture dynamic modulus make the prediction more complicated. In addition, it is expected that the laboratory ageing could be harder for MSCR *R3.2* due to its large variation.

It is evident that the change in binder property is not proportional to the change in dynamic modulus. Similarly, the change in dynamic modulus is not comparable to the change in the HWT rut depth. Therefore, it is expected that when different binder properties are applied to predict mixture performance, the shift factor would be varied greatly.

## 6. Conclusions

This paper evaluated the effect of field pavement ageing on properties of asphalt binder and asphalt mixture at two different rounds. The relationship between property changes in the asphalt and asphalt mixture due to ageing was also analyzed.

Results indicate that asphalt pavement in general became stiffer after years of service in terms of asphalt binder (high temperature PG, MSCR) and asphalt mixture (dynamic modulus, HWT rut depth). The asphalt mixture stiffening is caused to a large degree by asphalt aging, considering that the evaluated cores had no significant differences between the two rounds in in-place air voids, aggregate gradation and asphalt content. Such findings were confirmed by observing a good relationship between the change in binder properties and mixture properties. However, the application of the chip seal significantly reduced the aging process.

The material properties changed to different extents over time. The parameters that are most sensitive to field ageing are MSCR *R3.2* and dynamic modulus. Thus, the comparison among lab-tested material properties may be better checked in rank instead of absolute values. Note that the high variation in MSCR *R3.2* could cause some issues for laboratory aging, which needs further research. The effect of asphalt aging on pavement mixture properties may not follow a proportional liner trend.

The PG76-22 polymer-modified binders in general aged slower than other binder types evaluated in this study, for both asphalt binder and asphalt mixture properties. Since it is already proved that PG 76-22 polymer-modified binders show very good rutting resistance, such slow aging could also be beneficial to cracking resistance.

However, there are also some limitations that need improvement in future studies. For example, only two existing pavement structure types were selected, including flexible pavements, and a combination of asphalt/PCC pavements. For other types of asphalt pavement, the effects of field aging on material properties and rutting performance and their mechanisms need to be further studied.

## Figures and Tables

**Figure 1 materials-16-00225-f001:**
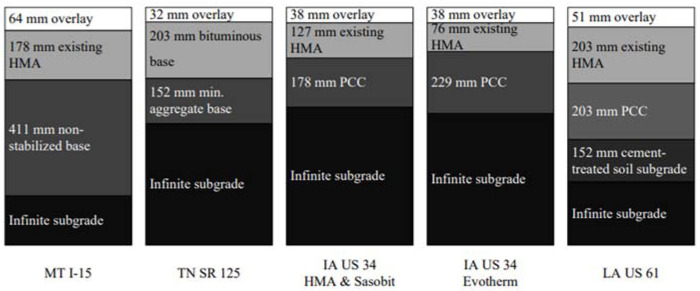
Pavement structure diagram of each project. (From top to bottom in turn are overlay, existing HMA, base, and subgrade respectively).

**Figure 2 materials-16-00225-f002:**
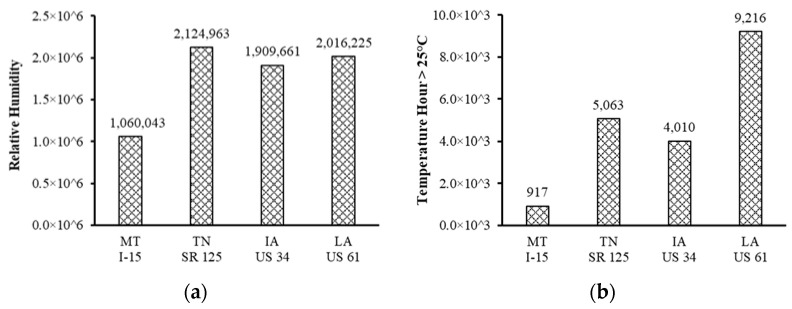
Field climate information: (**a**) relative humidity; (**b**) air temperature hour > 25 °C; (**c**) shortwave solar radiation.

**Figure 3 materials-16-00225-f003:**
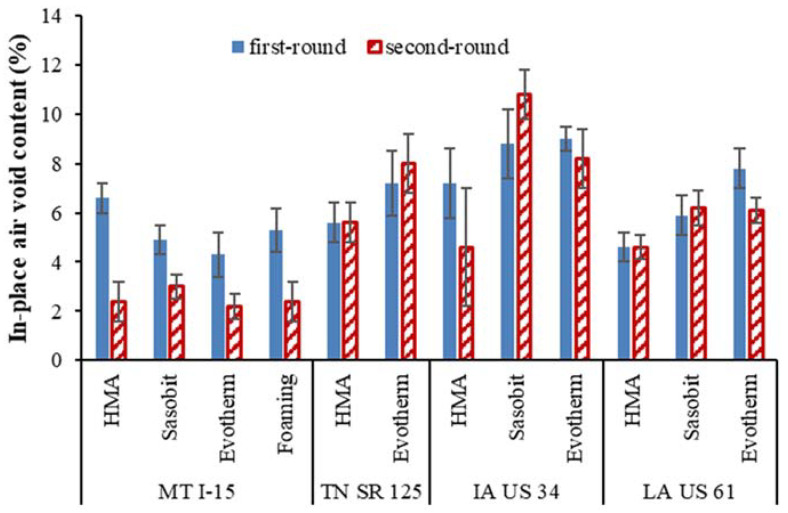
Comparison of in-place air voids.

**Figure 4 materials-16-00225-f004:**
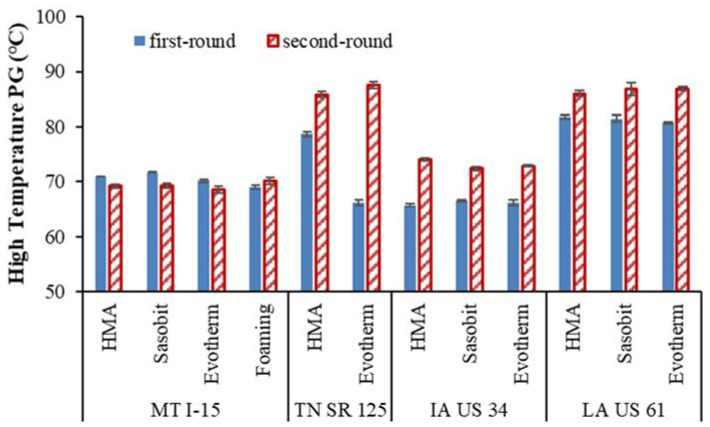
Comparisons of extracted binder high-temperature PGs.

**Figure 5 materials-16-00225-f005:**
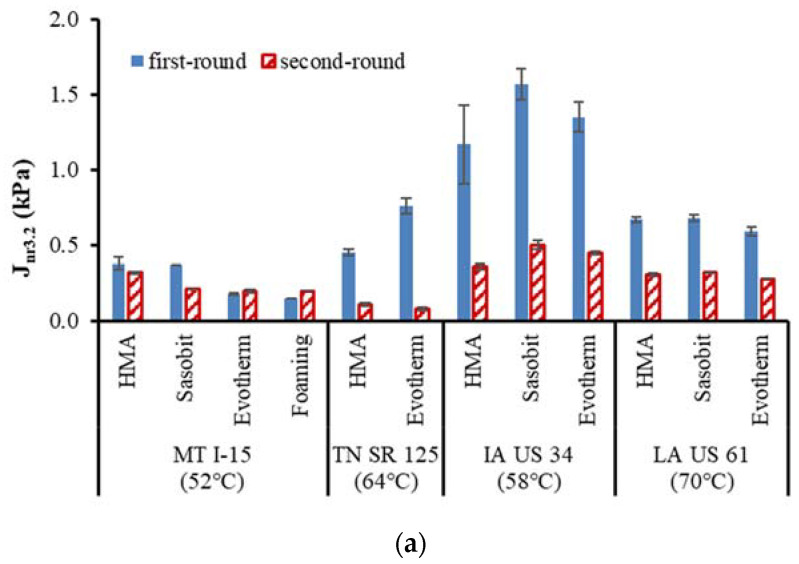
Comparisons of recovered binder MSCR results (**a**) *Jnr3.2*, and (**b**) *R3.2*. Note: the number shown in the parenthesis indicates test temperatures.

**Figure 6 materials-16-00225-f006:**
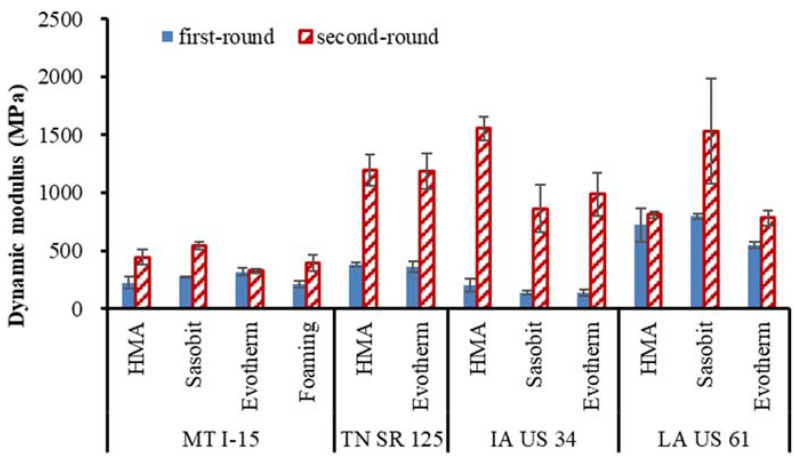
Comparisons of field core dynamic modulus at 30 °C and 0.1 Hz.

**Figure 7 materials-16-00225-f007:**
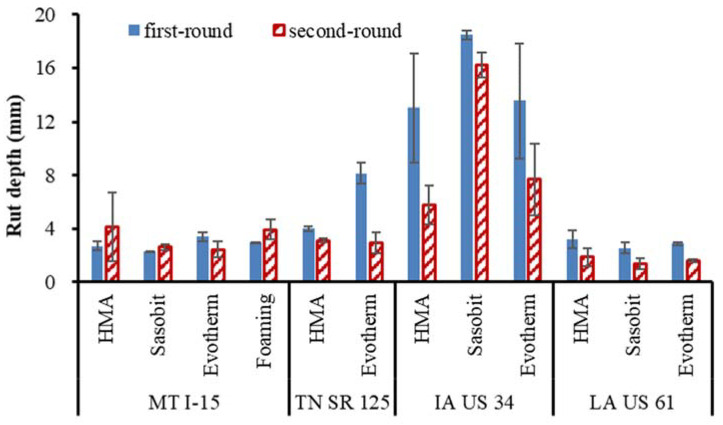
Comparisons of field core Hamburg rut depth.

**Figure 8 materials-16-00225-f008:**
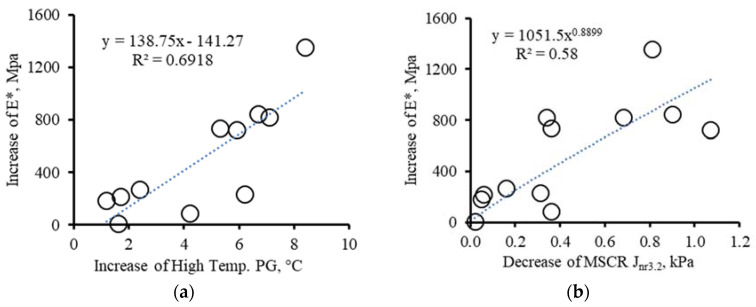
Effect of asphalt binder property change on dynamic modulus change (**a**) increase in high-temperature PG, (**b**) decrease in MSCR *Jnr3.2* and (**c**) increase in MSCR *R3.2*.

**Figure 9 materials-16-00225-f009:**
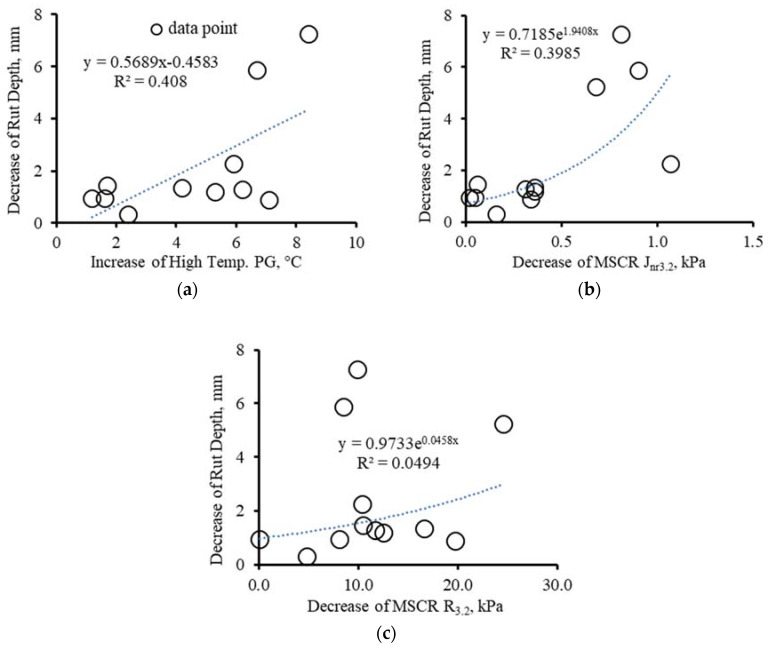
Effect of asphalt binder property change on HWT rut depth change (**a**) increase in high-temperature PG, (**b**) decrease in MSCR *Jnr3.2*, and (**c**) increase in MSCR *R3.2*.

**Figure 10 materials-16-00225-f010:**
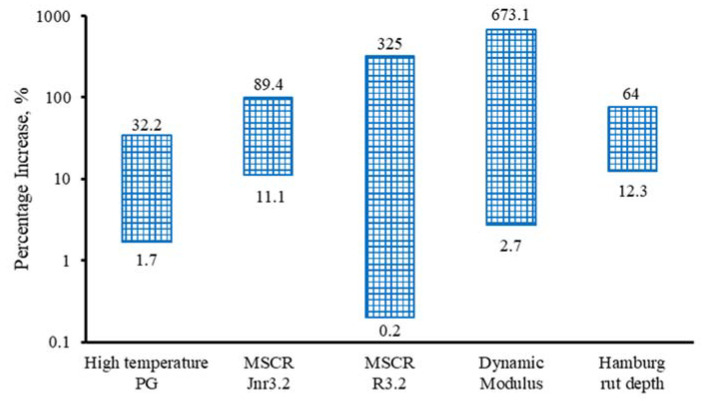
Percentage increase/decrease in each material property.

**Table 1 materials-16-00225-t001:** Specimen Grading for Indoor Test.

Parameter	MT I-15	TN SR 125	IA US 34	LA US 61
Construction year	2011	2011	2011	2012
Warm mix type	Sasobit, Evotherm DAT, Foam	Evotherm 3G	Sasobit, Evotherm 3G	Sasobit, Evotherm 3G
Mixing temp. (°C)	H (157–160) W (139–149)	H (160–177) W (143–160)	H (166–171) W (129–138)	H (163) W (146)
Compaction temp. (°C)	H (143–149) W (132–141)	H (155–168) W (135–143)	H (123–104) W (104–115)	H (136–158) W (118–121)
Traffic, AADTT	833	451	703	4779
Aggregate	Siliceous	Gravel and sand	Limestone, quartzite, sand	Granite and limestone
NMAS (mm)	19	12.5	12.5	12.5
Asphalt binder	PG 70-28	PG 70-22	PG 58-28	PG 76-22
Anti-stripping agent	hydrated lime, 1.4%	AZZ-MAZ, 0.3%	none	0.6%
Polymer modified	SBS	Yes	none	SBS
Asphalt content (%)	4.6	6.0	5.4	4.7
G_mm_	H (2.458) S (2.466) E (2.459) F (2.453)	H (2.352) E (2.355)	H (2.423) S (2.428) E (2.429)	H (2.464) S (2.468) E (2.464)
Sampling date, 1st round	Sep., 2011	Oct., 2011 and Aug.,2012	Sep., 2011	May–June, 2012
Sampling date, 2nd round	Aug., 2013	Dec., 2014	Dec., 2014	Feb., 2015
RAP or RAS	none	10% RAP	17% RAP	15% RAP

Note: H—HMA, W—WMA, S—Sasobit, E—Evotherm, F—Foaming, NMAS—nominal maximum aggregate size, AADTT—average annual daily truck traffic.

**Table 2 materials-16-00225-t002:** Summary of Laboratory Mixture and Binder Testing.

Test	IDT, Mixture	HWT, Mixture	DSR, Binder	DSR, Binder	Other Test
Test conditions	Temperature (°C): −20, −10, 0, 10, 20, 30 Frequency (Hz): 20, 10, 5, 1, 0.1, 0.01	50 °C	Temperature: depends on asphalt	High pavement temperature, stress: 0.1, 3.2 kPa	Depends on test
Material properties	Dynamic modulus	Rut depth	High temperature PG	MSCR J_nr_, MSCR R	AV, Gradation and AC
References/standards	Wen et al. 2002	AASHTO T 324	AASHTO MP 1/T 313/T 315	AASHTO T 350	AASHTO T308 AASHTO T30 AASHTO T209

Note: IDT—indirect tension test, HWT—Hamburg wheel track, DSR—dynamic shear rheometer, AMPT—asphalt mixture performance tester, PG—performance grade, MSCR—multiple stress recovery, Jnr—non-recoverable creep compliance, R—percent of recovery, AV—air voids, AC—asphalt content.

**Table 3 materials-16-00225-t003:** Aggregate Gradation Percent Passing Comparison Between the First- and the Second-Round Survey.

Sieve Size (mm)	MT I-15		TN SR 125		IA US 34		LA US 61	
	1st Round	2nd Round	1st Round	2nd Round	1st Round	2nd Round	1st Round	2nd Round
19.0	100	100	100	100	100	100	100	100
12.5	89–93	90–93	98	97–98	91	95–96	97–98	97–98
9.5	71–79	70–75	88–89	87–88	81	85–87	82–85	85–86
4.75	45–55	48–50	66–69	63–65	58–62	65–67	52–53	53–56
2.36	32–37	30–33	45	42–44	42–43	45–48	36	35–36
1.18	22–26	24	-	-	28–29	31–33	25–26	21–24
0.6	19–22	19	25–26	25–26	19	21–23	18–19	16–17
0.3	14–17	14–15	13	13	9–10	11–12	12	10–11
0.15	9–12	10	7–8	8	5–6	7	7–8	7
0.075	6–7	6–7	6	6	4–5	5	5	5

## Data Availability

All data, models, or codes that support the findings of this study are available from the corresponding authors upon reasonable request, including the field test data of the asphalt layers.

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
