# Peer review of "Effects of Field Aging on Material Properties and Rutting Performance of Asphalt Pavement"

_materials, 2022, doi:10.3390/ma16010225_

Round 1

Reviewer 1 Report

Wang et al. have presented the manuscript titled: Effects of Field Aging on Material Properties and Rutting Performance of Asphalt Pavement. Overall presentation of the article is good, but there require few modification before being publish, suggestions are as follow;

1.      Abstract should be strong as the potential of the work, authors should highlight the achieved results values in abstract like, Comparison of in-place air voids, Recovered binder MSCR test etc.

2.      In the introduction section, it is hard to find the existence of problem statement, why authors have performed this research work, what was lacking in previous studies which have compelled the authors to carry out this research.

3.      I would suggest he authors to revise the manuscript carefully to avoid any kind of spelling and grammatical mistakes.

4.      In the last sentence of abstract, please add a sentence which specify the impact of this study in the society.

Reviewer 2 Report

Dear authors, the article presents good results and is within the scope of the journal, however, several improvements need to be made, mainly in the abstract, introduction and discussion of the results.

The abstract is unattractive. I suggest that the authors can bring the main results and most important conclusions.

Introduction: The introduction is very succinct and also unattractive, it does not present a brief literature review that allows understanding in which the present work is contributing to the technical and scientific advance. It is not clear in the introduction to the innovation of the work. I recommend that the authors can detail the great contribution of the work.

Results: The results are well presented, however, there is not enough discussion in light of the literature. I recommend that the authors can explain the limitations of the present work, as well as the prospects for future research.
